# Effect of social and digital media mental health messaging on mental health help-seeking behaviors in the sub-Saharan African population: A systematic review protocol

Priscilla Aboagyewaa Boateng[1], Isaiah Osei Duah Junior[2], Josephine Ampong[3], Margaret Dowuona-Hammond[4], Peter J. Schulz[5,6]*, Laura Marciano[1,7]*

1 The Media School, Indiana University Bloomington, Bloomington, Indiana, United States of America, 2 Department of Psychology, College of Health and Human Sciences, North Carolina Agricultural and Technical State University, Greensboro, North Carolina, United States of America, 3 Department of Optometry and Visual Sciences, College of Science, Kwame Nkrumah University of Science and Technology, Kumasi, Ghana, 4 Department of Strategic Communication, Public Relations and Advertising, Universitat Autònoma de Barcelona, Barcelona, Spain, 5 Faculty of Communication, Culture, and Society, University of Lugano, Lugano, Switzerland, 6 Wee Kim Wee School of Communication and Information (WKWSCI), Nanyang Technological University (NTU), and Department of Communication & Media, Ewha Womans University, South Korea, 7 Digital Wellness Lab, Boston Children's Hospital, Boston, United States of America

* peter.schulz@usi.ch, laumarci@iu.edu

## Abstract

Mental health is a major public health concern, with disproportionate burdens in sub-Saharan Africa (SSA), where access to care is limited and stigma remains high. Social and digital platforms, including social media, mobile health (mHealth) applications, and SMS-based messaging, provide opportunities for information sharing, peer engagement, and tailored interventions that may enhance literacy and normalize help-seeking. Yet, they pose risks, including misinformation, exposure to harmful content, and reinforcement of stigma in diverse contexts. Despite this potential, evidence from SSA on the effects of social and digital media messaging on mental health knowledge, attitudes, behaviors, and help-seeking is scarce. This systematic review will assess the effect of social and digital media mental health messaging on help-seeking behaviors in SSA. Electronic databases, including PubMed, Psychological Information Database (PsycINFO), Cumulative Index to Nursing and Allied Health Literature (CINAHL), Communication and Mass Media Complete, Scopus, Cochrane Central Register of Controlled Trials (CENTRAL), Web of Science, Educational Resource Information Centre (ERIC), ProQuest Sociology, Google Scholar and Embase will be systematically queried using predefined keywords without language restrictions to ensure comprehensive evidence capture. Eligible studies will include interventions delivering mental health messaging through social media, mHealth applications, SMS, web-based platforms, or hybrid approaches, analyzing behavioral and psychological outcomes, and any kind of intervention studies.

**Data availability statement:** No datasets were generated or analysed during the current study. All relevant data from this study will be made available upon study completion.

**Funding:** The author(s) received no specific funding for this work.

**Competing interests:** The authors have declared that no competing interests exist.

**Abbreviation:** CENTRAL: Cochrane Central Register of Controlled Trials, CINAHL: Cumulative Index to Nursing and Allied Health Literature, DALYs: Disability-Adjusted Life Years, eHealth: Electronic Health, ERIC: Education Resources Information Center, GRADE: Grading of Recommendations Assessment, Development, and Evaluation, GRADE-CERQual: Grading of Recommendations Assessment, Development, and Evaluation: Confidence in the Evidence from Reviews of Qualitative Research, I²: I-squared statistic (measures between-study heterogeneity in meta-analysis), ID: Identifier, JBI: Joanna Briggs Institute, MEDLINE: Medical Literature Analysis and Retrieval System Online, mHealth: Mobile Health, NHLBI: National Heart, Lung, and Blood Institute, PICO: Population, Intervention, Comparator, Outcome, PICOS: Population, Intervention, Comparator, Outcomes, Study design, PRISMA: Preferred Reporting Items for Systematic Reviews and Meta-Analyses, PRISMA-P: Preferred Reporting Items for Systematic Review and Meta-Analysis Protocols, PROSPERO: International Prospective Register of Systematic Reviews, PsycINFO: American Psychological Association (APA) database, RoB 2: Cochrane Risk of Bias 2, ROBINS-I: Risk of Bias in Non-randomized Studies of Interventions, SMS: Short Message Service, SSA: Sub-Saharan Africa, STROBE: Strengthening the Reporting of Observational Studies in Epidemiology, YLDs: Years Lived with Disability

Methodological quality and risk of bias will be assessed using validated tools appropriate to each study design, including the Cochrane Risk of Bias 2 (RoB 2) tool, Risk of Bias in Non-Randomized Studies of Interventions (ROBINS-I), and Strengthening the Reporting of Observational Studies in Epidemiology (STROBE) checklist. Where appropriate, data will be synthesized with or without meta-analysis. This synthesis will clarify how social and digital media shape mental health outcomes, describe patterns across delivery channels, identify gaps, and inform culturally sensitive interventions to improve communication and promote mental health help-seeking in SSA.

## Introduction

Mental health is a major public health concern, with a notable resurgence during and after the coronavirus disease pandemic, yet access to appropriate care remains limited and disparities persist across regions [1–5]. Globally, one in eight people, approximately 970 million, live with a mental disorder [6,7], accounting for about 32.4% of years lived with disability (YLDs) and 13.0% of disability-adjusted life years (DALYs) worldwide [8]. The burden is particularly pronounced in sub-Saharan Africa (SSA), where prevalence estimates suggest that 3.3–9.8% of the population experiences mood disorders and 5.7–15.8% experiences anxiety disorders. Yet up to 85% of those affected receive no treatment [9]. This lack of appropriate mental health services creates ripple effects across communities, exacerbating untreated conditions, straining fragile health systems, and deepening social and economic disparities [10–16]. Access to professional mental health services remains severely constrained in the region, with the median workforce, including psychiatrists, nurses, psychologists, and social workers, at only 1.6 per 100,000 people, compared with a global median of more than 13 per 100,000 [17]. This paucity of services is further aggravated by stigma surrounding mental illness, which discourages individuals from seeking care and leaves many conditions untreated, increasing pressure on already fragile health systems [16]. Consequently, there is an urgent need to implement cost-effective and accessible mental health delivery systems that can reach underserved populations, reduce barriers to care, and strengthen the resilience of health infrastructures.

Of note, the advent of social and digital media platforms, including social media (Facebook, WhatsApp, X, YouTube, TikTok), mobile health (mHealth) applications, SMS-based messaging, and web-based platforms, has significantly transformed communication dynamics [18–23]. These platforms create new avenues for rapid information dissemination, peer engagement, and interactive dialogue [24–28]. Unlike traditional health communication channels, these digital platforms enable users not only to receive information but also to actively engage in discussions about mental health, share personal experiences, and seek guidance from peers and professionals [27]. By harnessing the extensive reach of social and digital media, individuals can access mental health resources and peer support networks in ways that traditional systems often cannot, overcoming geographic, social, and economic barriers to care [29,30]. While social and digital media usage can accelerate mental health literacy,

promote open communication, and encourage help-seeking behaviors by normalizing conversations around psychological well-being, it also offers opportunities for tailored interventions [29,31–33]. For instance, social media platforms enable targeted awareness campaigns [30], psychoeducational content [34], and peer-led support groups [35] adapted to diverse populations [30, 34,35]. Their interactive nature allows for real-time feedback, helping health communicators to evaluate engagement [36], track behavioral responses [37,38], and refine messaging strategies to maximize effectiveness and reach [36–38]. However, these benefits are accompanied by potential drawbacks, including the spread of misinformation, exposure to negative or triggering content, and unequal access, which may exacerbate existing disparities in mental health support [39–44].

Furthermore, the potential of social media to shape mental health perceptions is particularly significant, especially in SSA, which has an estimated 384 million users, a figure that continues to grow with expanding internet connectivity and mobile penetration [45]. WhatsApp and Facebook are among the most widely used platforms across the region, with Instagram, YouTube, and TikTok also gaining substantial traction, particularly among youth populations [46]. This uptake unfolds within diverse cultural contexts where traditional healing practices, religious beliefs, and community-based support systems intersect with modern healthcare [47,48]. These dynamics strongly influence how mental health messages on social media are interpreted, trusted, or resisted, making it essential to align messaging strategies with local cultural realities [49]. Positive narratives on these platforms can improve attitudes toward services and encourage help-seeking [29,50]; however, poorly framed or misleading content may reinforce harmful stereotypes or spread misinformation [51,52]. Indeed, some stigma-reduction campaigns, when inappropriately designed, have unintentionally deepened negative perceptions [53–55]. For instance, fear-based HIV campaigns in South Africa, while aiming to promote safer behaviors, reinforced stigma toward people living with HIV rather than reducing it [56]. Similarly, a study in Nigeria shows that framing mental illness in terms of dangerousness or supernatural explanations can worsen public stigma and discourage care-seeking [57]. These insights highlight the importance of considering not only the extent of social media use but also the design, framing, and reception of messages [58,59]. Consistent with these views, framing theory suggests that the way messages are presented influences how audiences interpret and respond [60], while narrative persuasion theory emphasizes how stories and testimonies facilitate empathy, identification, and behavioral change [61]. Yet when messages are misaligned with cultural expectations, they can yield unintended consequences such as reinforcing stigma or mistrust [47,62,63]. In SSA, where cultural and community-based frameworks strongly shape mental health perceptions [49,64], these theories are especially relevant for ensuring that social media campaigns encourage help-seeking rather than harm. Recognizing these dynamics, particularly the interaction between framing, narratives, and cultural expectations, is therefore crucial for developing culturally sensitive digital interventions [48,65].

Despite the relevance of social and digital media to shape mental health awareness and help-seeking, evidence from SSA remains scarce. Existing studies and/or systematic reviews have largely emphasized prevalence [10,66–68], barriers to care [69,70], focused on Western contexts [71], narrow groups, or specific interventions such as mobile apps and SMS campaigns [72–74]. In SSA, few studies have examined how social and digital media messaging itself, through its design, audience characteristics, or unintended effects, influences mental health knowledge, attitudes, behaviors, and help-seeking [50,71,75]. This systematic review aims to (1) map the use of social and digital media–based mental health messaging in SSA, including social media platforms, SMS-based interventions, web-based platforms, and hybrid approaches, and (2) evaluate the impact of these messages on psychological and behavioral outcomes, with particular attention to help-seeking behaviors. The findings will provide evidence to guide the development of culturally relevant digital interventions and strengthen strategies for mental health communication in SSA.

## Materials and methods

The current review protocol, registered in the International Prospective Register for Systematic Reviews (PROSPERO ID: CRD420251151310), has been prepared following the PRISMA-P checklist (S1 Table) to ensure methodological rigor and



transparency [76,77]. The final review will adhere to PRISMA guidelines [78,79] and the Cochrane Handbook to ensure robustness and reproducibility [80]. This review aims to synthesize evidence on the effect of mental health messaging on social and digital media platforms (e.g., WhatsApp, Facebook, Instagram, Twitter/X, YouTube, TikTok, mobile health applications, SMS-based messaging, web-based platforms) on outcomes such as knowledge (e.g., awareness of symptoms and services), attitudes (e.g., reductions in stigma), empowerment (e.g., confidence in managing stress), behaviors (e.g., adoption of coping strategies), and help-seeking (e.g., intention or action to consult a professional) among populations in SSA. Messaging examples may include Facebook campaigns, WhatsApp-based health groups, or TikTok videos promoting resilience, mobile app notifications with mental health tips, SMS-based psychoeducation programs, or web-based mental health education modules. Online communities and peer-support forums will be included if the mental health messaging is disseminated via social media or other digital platforms included in this review, but excluded if they exist only as stand-alone private forums without broader digital messaging components.

Additionally, where studies include comparators, it will examine differences in outcomes between individuals exposed to mental health messaging via social and digital media and those with no exposure or alternative messaging approaches, including variations in framing, content type, or platform (more information about specific interventions and outcomes can be found in Table 1). The protocol outlines the planned methodology, including the search strategy, study selection criteria, data extraction, risk of bias assessment, and synthesis approach. Study selection and database screening will be systematically documented using a PRISMA flow diagram (Fig 1). By following a pre-registered protocol, this review ensures that

**Table 1. Search terms adapted for all databases.**

| PICOS Element | Description/Search Terms |
|---|---|
| **Population (P)** | Individuals in SSA, all countries in SSA, the region as a whole, continental Africa, or general African populations |
| **Intervention/ Exposure (I)** | Social media or digital communication: social media, social networks, digital platforms, online platforms, internet, internet-based, web-based, online communication, online community, online network, digital communication, virtual platforms, cyberspace, mobile health (mHealth), eHealth, telehealth, telemedicine, Facebook, WhatsApp, Instagram, YouTube, TikTok, X, Messenger, Telegram, Snapchat, LinkedIn, WeChat, Viber, IMO, Ayoba, Share-Chat, Pinterest, Reddit, Threads, Clubhouse, Google+, MySpace<br>combined with mental health–related terms such as mental health, mental illness, well-being, psychological well-being, psychological distress, psychiatric disorder, mental disorder, behavioral health, major depressive disorder, dysthymia, bipolar disorder, panic disorder, social anxiety disorder, obsessive-compulsive disorder, post-traumatic stress disorder, PTSD, acute stress disorder, adjustment disorder, eating disorder, anorexia nervosa, bulimia nervosa, binge eating disorder, schizophrenia, schizoaffective disorder, psychosis, psychotic disorder, delusion, personality disorder, borderline personality disorder, antisocial personality disorder, narcissistic personality disorder, attention deficit hyperactivity disorder, ADHD, autism spectrum disorder, substance use disorder, alcohol use disorder, drug use disorder, addiction, gambling disorder, sleep disorder, insomnia, circadian rhythm disorder, burnout, stress, suicidal ideation, suicide, self-harm, self-esteem, mindfulness, emotional regulation, loneliness, social support, quality of life, life satisfaction, happiness, flourish, positive psychology |
| **Comparison (C)** | No exposure, alternative messaging approaches, or standard practice (if applicable) |
| **Outcomes (O)** | **Primary:** Help-seeking (intentions to seek professional support, use of formal mental health services, engagement with informal support networks) **Secondary:** Mental health knowledge (mental health disorders, treatment options, prevention, coping strategies, self-care), attitudes (stigma reduction, acceptance, openness, positive perceptions of help-seeking), behaviors (self-care, treatment adherence, lifestyle changes), empowerment (perceived ability to manage mental health, confidence, self-efficacy), engagement metrics (message reach, interaction rates, self-reported engagement) |
| **Study Design (S)** | All study designs; no restrictions on language. Searches will include controlled vocabulary and free-text terms; Boolean operators and truncation will be applied; strategies adapted per database (PubMed, PsycINFO, CINAHL, CMMC, Scopus, WoS, ERIC, ProQuest Soc., Medline (ProQuest), Embase) |

PubMed, PsycINFO: Psychological Information Database, CENTRAL: Cochrane Central Register of Controlled Trials, CINAHL: Cumulative Index to Nursing and Allied Health Literature, CMMC: Communication & Mass Media Complete, Scopus: Scopus, WoS: Web of Science Core Collection, ERIC: Education Resources Information Center, ProQuest Soc.: ProQuest Sociology, Medline (ProQuest): MEDLINE via ProQuest, Embase: Excerpta Medica Database.

## Identification of studies via databases and registers

**Identification**

Records identified from databases (n = )

- Communication & Mass Media Complete (n = )
- Psychology and Behavioral Sciences Collection (n = )
- Educational Resource Information Center (n = )
- Web of Science (n = )
- PsycINFO (n = )
- CINAHL (n = )
- Medline (PROQUEST) (n = )
- ProQuest Sociology (n = )
- PubMed (n = )
- Embase (n = )
- CENTRAL and/or Cochrane Library (n = )
- Google Scholar (n = )

Records removed *before screening*:
- Duplicate records removed (n = )
- Records marked as ineligible by automation tools (n = )
- Records removed for other reasons (n = )

**Screening**

Records screened (n = ) → Records excluded (n = )

Reports sought for retrieval (n = ) → Reports not retrieved (n = )

Reports assessed for eligibility (n = ) → Reports excluded:
Reason 1 (n = )
Reason 2 (n = )
Reason 3 (n = )
etc.

**Included**

Studies included in review (n = )

**Fig 1. PRISMA flow diagram illustrating the study selection process.**

methodological decisions are pre-specified, reducing the risk of reporting bias and enhancing transparency in the evaluation of social and digital media interventions for mental health in SSA. This study is a protocol using secondary sources; hence, ethical approval was not required.

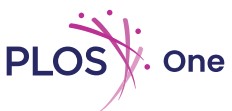

**Eligibility criteria**

**Inclusion.** The criteria for inclusion will follow the PICOS elements, which comprise the population, intervention, comparators, outcomes, and study design.

**Population (P).** Studies will be eligible if they examine populations residing in SSA, including adolescents, young adults, adults, and older adults. Countries considered within the review include Angola, Benin, Botswana, Burkina Faso, Burundi, Cabo Verde, Cameroon, Central African Republic, Chad, Comoros, Congo, Côte d'Ivoire, Democratic Republic of the Congo, Djibouti, Equatorial Guinea, Eritrea, Eswatini, Ethiopia, Gabon, Gambia, Ghana, Guinea, Guinea-Bissau, Kenya, Lesotho, Liberia, Madagascar, Malawi, Mali, Mauritania, Mauritius, Mozambique, Namibia, Niger, Nigeria, Rwanda, São Tomé and Príncipe, Senegal, Seychelles, Sierra Leone, Somalia, South Africa, South Sudan, Sudan, Tanzania, Togo, Uganda, Zambia, and Zimbabwe. Studies including multiple regions will be eligible if results specific to sub-Saharan African populations are reported separately.

**Intervention (I).** The intervention of interest is mental health messaging delivered via social and digital media platforms. These include campaigns, posts, videos, advertisements, educational content, digital storytelling, narrative interventions, notifications, SMS messages, or other content aimed at improving mental health literacy, reducing stigma, promoting empowerment, or encouraging help-seeking behaviors. Eligible social media platforms include Facebook, WhatsApp, X, Instagram, TikTok, YouTube, Snapchat, Pinterest, LinkedIn, Telegram, Messenger, WeChat, Viber, IMO, Ayoba, ShareChat, Reddit, Threads, Clubhouse, Google+, and MySpace. and similar platforms. Examples include Facebook mental health awareness campaigns, WhatsApp peer support groups, Instagram mental health education posts, YouTube psychoeducation videos and TikTok destigmatization content. Other digital media platforms include mobile health applications in standalone or integrated mobile applications delivering mental health messaging, educational content, notifications, reminders, or interactive features designed to improve mental health knowledge, attitudes, or behaviors. This includes mental health-specific apps, wellness apps with mental health components, and general health apps delivering mental health content. SMS-based interventions include text messaging services delivering mental health-related information, reminders, supportive messages, or behavior change content via SMS or similar mobile messaging protocols. Examples include SMS campaigns promoting mental help-seeking, text-based mental health education programs, or supportive text messages for individuals at risk. Web-based platforms include internet-based platforms, websites, portals, or online modules that deliver mental health messaging through interactive or static content. Email-based interventions include electronic mail campaigns, newsletters, or structured email programs delivering mental health content to the target population. Hybrid or multi-channel approaches include combining digital delivery channels (e.g., social media campaign in addition to SMS reminders, mHealth app in addition to WhatsApp support group, web platform in addition to email). Messaging may vary in format, frequency, duration, and framing, including persuasive communication, narrative storytelling, risk communication, or public health education. Exposure to mental health messaging will be operationalized based on how each included study measures it. Acceptable exposure measures include self-reported exposure, platform-generated metrics (reach, views, engagement), study-controlled exposure (randomized delivery, experimental manipulation), or behavioral indicators (click-through rates, time spent, sharing).

**Comparator (C).** When available, comparators include populations with no exposure to mental health messaging or exposure to alternative messaging approaches, such as different message framing, delivery platforms, content types, or campaign intensity.

**Outcome (O).** The primary outcome is objectively measured help-seeking behavior, defined as documented utilization of formal mental health services (e.g., verified clinic visits, contact with mental health professionals, enrollment in treatment programs, or validated self-reported service use). Secondary outcomes include help-seeking intentions (measured through validated scales), informal help-seeking (engagement with family, friends, or community support), mental health knowledge (understanding of disorders, treatments, and coping strategies), attitudes (stigma reduction, acceptance, openness to discussion), behavioral changes (self-care engagement, treatment adherence, lifestyle

modifications), empowerment (self-efficacy, confidence in seeking support), and engagement metrics (message reach, interaction rates, content exposure).

**Study design (S).** Eligible study designs include randomized controlled trials, cluster-randomized trials, quasi-experimental studies such as pre-post designs or interrupted time series, cohort studies, cross-sectional surveys, qualitative studies (e.g., interviews, focus groups, ethnographies) and mixed-methods studies examining exposure to social media mental health messaging. Both quantitative and qualitative outcomes related to mental health messaging will be included.

**Exclusion.** Studies not reporting relevant mental health outcomes, not conducted in SSA, or focusing solely on offline interventions will be excluded. Reviews, editorials, commentaries, and conference abstracts without full-text data will be excluded. Additionally, studies examining purely synchronous counseling without messaging, in-person interventions with minimal digital components, electronic health records without patient messaging, or wearable devices without messaging features will be excluded.

## Data sources

Electronic databases to be systematically searched include PubMed, PsycINFO (Psychological Information Database), CENTRAL (Cochrane Central Register of Controlled Trials), CINAHL (Cumulative Index to Nursing and Allied Health Literature), Communication & Mass Media Complete, Scopus, Web of Science Core Collection, ERIC, ProQuest Sociology, Psychology and Behavioral Sciences and Embase. Searches will cover studies from January 2000 to September 2025. In addition, hand searches will be conducted on the first ten pages of Google Scholar, and reference lists of included studies and relevant reviews will be reviewed manually. Key journals focusing on global mental health, digital health interventions, and social media studies will also be reviewed for potentially relevant studies not indexed in the electronic databases.

## Search terms and search strategies for identifying studies

Searches will be conducted in the aforementioned databases using four main concepts aligned with the PICOS framework (Table 1): mental health terms, messaging or communication terms, social media or digital terms, and population terms. Without language restriction, we will conduct an extensive search for primary studies published between January 2000 and September 2025, a period that reflects the rise and widespread use of social media. Boolean operators and truncation will be applied to combine synonyms and variations, and the strategy will be adapted for each database using controlled vocabulary and free-text terms. Mental health terms include, but are not limited to, mental health, mental illness, wellbeing, psychological well-being, psychological distress, psychiatric disorders, affect, emotion, depression, major depressive disorder, dysthymia, bipolar disorder, anxiety, panic disorder, phobias, social anxiety disorder, obsessive-compulsive disorder, post-traumatic stress disorder, acute stress disorder, adjustment disorder, eating disorders, schizophrenia, psychosis, personality disorders, attention deficit hyperactivity disorder, autism spectrum disorders, substance use disorders, alcohol and drug use disorders, addiction, gambling disorder, sleep disorders, burnout, stress, suicidal ideation, self-harm, resilience, coping, self-esteem, mindfulness, emotional regulation, loneliness, social support, quality of life, life satisfaction, happiness, flourishing, and positive psychology. Messaging or communication terms include message, communication, health promotion, health campaign, public health message, media message, mental health communication, intervention, health education, information, content, outreach, digital health communication, risk communication, social marketing, online message, persuasive communication, narrative, storytelling, framing, health information, information dissemination, knowledge translation, and education. Social and digital media terms include social media, social networks, digital platforms, online platforms, internet, internet-based, web-based, online communication, online community, online network, digital communication, virtual platforms, cyberspace, mobile health, mHealth, eHealth, telehealth, telemedicine, Facebook, WhatsApp, Instagram, YouTube, TikTok, X, Messenger, Telegram, Snapchat, LinkedIn, WeChat, Viber, IMO, Ayoba, ShareChat, Pinterest, Reddit, Threads, Clubhouse, Google +, and MySpace. Population terms include all countries in

SSA, the region as a whole, continental Africa, or general African populations. Full search strategies will be documented in the review report for transparency and reproducibility.

## Study selection

All retrieved records will be exported to EndNote reference management software for automatic duplicate exclusion. Subsequently, the references will be exported to the Covidence platform for titles, abstract screening, and full text review. For non-English studies, initial screening of titles and abstracts will be conducted using verified machine translation tools (e.g., Google Translate, DeepL), with cross-verification by bilingual team members where available. For full-text articles in non-English languages, we will prioritize professional translation services when feasible, with particular attention to articles that will be included in the final review. Bilingual team members with expertise in health research will verify translations of key methodological details, including study design, population characteristics, intervention components, and outcome measures. To ensure accurate translation of nuanced concepts such as mental health, help-seeking, stigma, and culturally-specific terminology, we will: (1) maintain a translation glossary documenting how key terms are rendered across languages, (2) consult with study authors when terminology is ambiguous or lacks direct equivalents, (3) document translation decisions and uncertainties in our data extraction forms, and (4) note any limitations related to translation in the final review. Where critical details remain unclear after translation, we will contact the study authors for clarification. Specifically, two independent reviewers (P.A.B. and J.A.) will screen titles and abstracts for eligibility based on the a priori eligibility criteria. A Cohen's kappa will be calculated as a measure of intercoder reliability. Discrepancies will be resolved through discussion, and a third reviewer will adjudicate unresolved disagreements. Full-text articles will then be screened independently by two reviewers according to the pre-specified inclusion and exclusion criteria, and reasons for exclusion at the full-text stage will be documented in a PRISMA flow diagram.

## Data extraction

Data will be extracted using a standardized form developed and piloted prior to full extraction. Extracted information will include study characteristics such as author, year, country, study design, and setting; population characteristics including sample size, age, gender, urban or rural context, socioeconomic status, and other relevant information (e.g., community sample, clinical sample, etc.); intervention characteristics including social media platform, message type, duration, frequency, framing, and content type; comparator characteristics; outcomes including type, conceptualization, measurement tool, timepoints, and reported effects; key findings, including quantitative results (and, when possible, effect sizes and confidence intervals) and qualitative findings. For studies published in languages other than English, data will be extracted from full-text articles that have either been translated using language translation software or translated by bilingual team members with expertise in the subject. A second reviewer will verify the accuracy of the translations whenever possible, and any uncertainties will be documented and, if needed, clarified with the study authors. Two reviewers (P.A.B. and J.A.) will extract data independently, and discrepancies will be resolved through consensus.

## Risk of bias

The methodological quality and risk of bias of included studies will be assessed independently by two reviewers using tools appropriate to each study design. Randomized controlled trials will be evaluated using the Cochrane Risk of Bias 2 (RoB 2) tool. Non-randomized intervention studies (including quasi-experimental designs, pre-post studies, and interrupted time series) will be assessed using the Risk of Bias in Non-randomized Studies of Interventions (ROBINS-I) tool. The STROBE (Strengthening the Reporting of Observational Studies in Epidemiology) checklist will be used to assess reporting quality as a complement to bias assessment. Discrepancies will be resolved through discussion or consultation with a third reviewer.



## Quality assessment

To provide a comprehensive evaluation of study quality, the National Heart, Lung, and Blood Institute (NHLBI) Study Quality Assessment Tools will be employed for all included studies. These tools are specifically designed to assess internal validity, potential sources of bias, and overall methodological rigor across different study designs, including controlled intervention studies, cohort and cross-sectional studies, case-control studies, and before-after studies. Each study will be rated as good, fair, or poor quality based on criteria such as clarity of research question, adequacy of study population and sample size, appropriateness of exposure or intervention measurement, outcome assessment, statistical analyses, and control for confounding variables. The combination of design-specific risk of bias assessment tools (RoB 2, ROBINS-I) and the NHLBI quality appraisal ensures a thorough evaluation of the credibility and reliability of the evidence. Discrepancies in quality assessment will be resolved through discussion or consultation with a third reviewer (I.O.D.J. or L.M), and the quality ratings will be incorporated into the narrative synthesis and interpretation of findings.

## Data synthesis

For heterogeneous data, we plan to narratively consolidate the evidence by synthesis without meta-analysis [81]. Specifically, the narrative synthesis will be conducted due to anticipated heterogeneity in study designs, intervention types, platforms, and outcomes. The synthesis will be structured around population characteristics, intervention characteristics including platform, messaging type, duration, and framing, comparator type, and mental health outcomes including knowledge, attitudes, behaviors, empowerment, and help-seeking [81]. Where feasible, meta-analysis will be conducted using random-effects models if at least two studies are sufficiently homogeneous in terms of population, intervention, and outcomes. The *metaprop function* in R (from the *meta* package) will be used for pooling proportions and estimating overall effects. [82]. Effect sizes, including standardized mean differences or odds ratios, will be calculated with 95% confidence intervals. Statistical heterogeneity will be assessed using the $I^2$ statistic, with values of 25 percent (%), 50%, and 75% interpreted as low, moderate, and high heterogeneity, respectively. Subgroup analyses may be conducted based on age group, social media platform, message framing, or urban versus rural setting. Sensitivity analyses will be conducted by excluding studies with a high risk of bias.

To address anticipated heterogeneity in outcome measurement across diverse instruments (e.g., depression measured via PHQ-9, BDI, CESD, HADS), we will calculate standardized mean differences (SMD) to pool results across different scales measuring the same construct. Subgroup analyses will compare effect sizes by measurement type (validated vs. non-validated scales; Western-developed vs. culturally adapted instruments). If substantial heterogeneity persists ($I^2 > 75\%$), results will be presented separately by measurement approach. Narrative synthesis will follow Synthesis Without Meta-analysis (SWiM) guidelines, presenting structured tables organized by intervention type and outcome, effect direction plots, and textual synthesis identifying patterns in intervention characteristics and contextual factors. For qualitative or mixed-methods studies, we will extract and thematically synthesize findings related to acceptability, feasibility, barriers, facilitators, and mechanisms. Qualitative findings will be integrated with quantitative results to explain heterogeneity and provide a comprehensive understanding of intervention effects in sub-Saharan African contexts.

## Publication bias

Publication bias will be assessed through visual inspection of funnel plots if at least ten studies are included in a meta-analysis. Egger's test or Begg's test may also be conducted to quantitatively assess the presence of small-study effects.

## Confidence in cumulative evidence

The Grading of Recommendations Assessment, Development, and Evaluation (GRADE) approach will be used to assess the certainty of evidence across studies for each quantitative outcome. This involves evaluating the body of evidence

across five domains: risk of bias, inconsistency of results, indirectness, imprecision, and publication bias. Based on these domains, evidence will be rated as high, moderate, low, or very low based on considerations of risk of bias, consistency, directness, precision, and publication bias. For qualitative evidence, we will apply the GRADE-CERQual (Confidence in the Evidence from Reviews of Qualitative research) approach to assess confidence in qualitative findings, evaluating methodological limitations, coherence, data adequacy, and relevance. Applying GRADE and GRADE-CERQual will ensure that the review not only summarizes findings but also conveys the strength and reliability of the evidence, which is critical for informing culturally relevant interventions in SSA.

### Handling of missing data

In instances where included studies have missing, incomplete, or unclear data, efforts will be made to contact the corresponding authors to request additional information or clarification. When missing data cannot be obtained, the extent and nature of missing information will be documented and reported. For quantitative outcomes, the potential impact of missing data on study findings will be considered, and sensitivity analyses will be conducted when feasible to assess how assumptions about missing data affect the results. Studies with substantial or critical missing data that could compromise the validity of outcomes may be rated as lower quality or excluded from meta-analysis, while their findings will still be included in the narrative synthesis. The presence of missing data will be taken into account during the risk of bias assessment and quality appraisal using the Cochrane Risk of Bias 2 tool, ROBINS-I, and STROBE checklist. By systematically documenting and addressing missing data, the review aims to minimize potential bias and ensure that conclusions are based on the most reliable and complete evidence available.

### Study status and timeline

At the time of submission (October 2025), the systematic review had not yet commenced data collection or analysis. Database searches, screening, and data extraction will be conducted following the pre-registered protocol on PROSPERO. Data collection is scheduled to begin in November 2025, with completion expected by December 2025. Data synthesis and manuscript preparation are anticipated to be finalized by January 2026. No results have been generated to date.

### Ethics and dissemination

As this review will synthesize evidence from previously published studies, ethical approval is not required. Findings will be disseminated through peer-reviewed publication, presentations at international conferences, and policy briefs aimed at stakeholders involved in mental health promotion in SSA.

## Discussion

The proposed systematic review addresses a critical knowledge gap in understanding the role of social and digital media in shaping mental health awareness, attitudes, and help-seeking behaviors in SSA. While social and digital media platforms have been shown to facilitate rapid information dissemination, peer engagement, and interactive support globally, evidence from SSA remains limited. Most existing studies in the region have primarily focused on mental health prevalence [10,66–68], barriers to care [69,70], or the evaluation of specific digital interventions such as mobile applications and SMS campaigns [72,83] in Western countries. Few studies have examined how the design, framing, and audience characteristics of social media messaging directly influence mental health literacy, stigma reduction, and help-seeking behaviors in SSA [71,75, 84, 85]. By synthesizing available evidence, this review aims to clarify how social and digital media messaging can be leveraged to promote culturally sensitive and effective mental health interventions, particularly in a context where traditional support systems and community norms intersect with modern healthcare delivery. A key methodological strength of this review is the absence of language restrictions, which enhances comprehensiveness and reduces selection

bias. Given the linguistic diversity of SSA, this inclusive approach ensures that evidence from Francophone, Lusophone, Arabic-speaking, and other linguistic contexts is not excluded from the synthesis.

Despite its potential contributions, several limitations are anticipated in this review. First, the heterogeneity of study designs, social and digital media platforms, messaging formats, and outcome measures may limit the ability to draw generalized conclusions. Second, the rapidly evolving nature of social media use and digital health communication in SSA means that findings may quickly become outdated. Third, potential publication bias and the limited availability of peer-reviewed studies from low-resource settings could restrict the comprehensiveness of the evidence base. Finally, the review may be constrained by the quality and reporting of included studies, particularly regarding cultural adaptation, message framing, and the measurement of behavioral outcomes. Nonetheless, by systematically identifying and evaluating the current literature, this review will provide critical insights into digital mental health communication strategies and inform the design of interventions that are both evidence-based and culturally relevant for SSA populations.

## Supporting information

**S1 Table. PRISMA-P (Preferred Reporting Items for Systematic Review and Meta-Analysis Protocols) 2015 checklist: Recommended items to address in a systematic review protocol.**
(DOCX)

## Author contributions

**Conceptualization:** Priscilla Aboagyewaa Boateng, Isaiah Osei Duah Junior, Margaret Dowuona-Hammond, Laura Marciano.

**Data curation:** Priscilla Aboagyewaa Boateng.

**Formal analysis:** Isaiah Osei Duah Junior.

**Investigation:** Priscilla Aboagyewaa Boateng, Laura Marciano.

**Methodology:** Priscilla Aboagyewaa Boateng, Isaiah Osei Duah Junior, Josephine Ampong, Margaret Dowuona-Hammond, Laura Marciano.

**Project administration:** Priscilla Aboagyewaa Boateng, Isaiah Osei Duah Junior, Laura Marciano.

**Supervision:** Priscilla Aboagyewaa Boateng, Peter J. Schulz.

**Writing – original draft:** Priscilla Aboagyewaa Boateng, Isaiah Osei Duah Junior, Josephine Ampong, Margaret Dowuona-Hammond, Laura Marciano.

**Writing – review & editing:** Priscilla Aboagyewaa Boateng, Isaiah Osei Duah Junior, Josephine Ampong, Margaret Dowuona-Hammond, Peter J. Schulz, Laura Marciano.

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
