## [Decision Letter · Decision Letter 0]

9 Dec 2025

Dear Dr. Marciano,

Thank you for submitting your manuscript to PLOS ONE. After careful consideration, we feel that it has merit but does not fully meet PLOS ONE’s publication criteria as it currently stands. Therefore, we invite you to submit a revised version of the manuscript that addresses the points raised during the review process.

We look forward to receiving your revised manuscript.

Kind regards,

Olushayo Oluseun Olu

Academic Editor

PLOS One

Journal Requirements:

Reviewers' comments:

Reviewer's Responses to Questions

**Comments to the Author**

1. Does the manuscript provide a valid rationale for the proposed study, with clearly identified and justified research questions?

Reviewer #1: Yes

Reviewer #2: Yes

2. Is the protocol technically sound and planned in a manner that will lead to a meaningful outcome and allow testing the stated hypotheses?

Reviewer #1: Partly

Reviewer #2: Partly

3. Is the methodology feasible and described in sufficient detail to allow the work to be replicable?

Reviewer #1: Yes

Reviewer #2: Yes

4. Have the authors described where all data underlying the findings will be made available when the study is complete?

Reviewer #1: Yes

Reviewer #2: No

5. Is the manuscript presented in an intelligible fashion and written in standard English?

Reviewer #1: Yes

Reviewer #2: Yes

You may also provide optional suggestions and comments to authors that they might find helpful in planning their study.

Reviewer #1: 1. Strengths

This protocol addresses an important and well-justified research question with clear public health relevance for sub-Saharan Africa. The authors demonstrate strong methodological planning through PROSPERO registration and adherence to PRISMA-P, PRISMA, and Cochrane Handbook guidance. The proposed search strategy is broad and inclusive, covering multiple databases without language restrictions, and the inclusion of both quantitative and qualitative evidence is appropriate for the topic. Furthermore, the plan to apply GRADE for assessing certainty of evidence reflects a commitment to rigorous synthesis and transparent reporting. However, the protocol could benefit from the following clarifications and improvements:

2. Line Numbers

The authors should ensure that the manuscript includes numbered lines in all subsequent submissions. Line numbering is essential for reviewers to reference specific sections accurately and provide detailed feedback.

3. Primary Outcome

The primary outcome (p.g.15), “help-seeking,” is not operationalized with sufficient precision. It currently includes intentions, formal service utilization, and engagement with informal networks, which are conceptually distinct (intention vs action; formal vs informal). Without clear definitions and prioritization, synthesis will be challenging and may bias interpretation.

The authors could provide explicit operational definitions and a hierarchy for help-seeking outcomes (e.g., primary outcome = objectively measured help-seeking behavior such as service utilization; secondary outcomes = help-seeking intentions, informal help-seeking, self-efficacy). These clarifications will improve consistency and interpretability of results.

4. Exposure (Messaging) Measurement and Comparators

The definition and measurement of exposure to “mental health messaging” require greater clarity. Currently, the term encompasses campaigns, posts, apps, SMS, and peer groups, but the protocol does not specify how exposure will be measured (e.g., self-report, platform analytics, dose/frequency) or whether message content will be coded systematically. Without clear exposure metrics, comparability across studies will be limited.

5. Risk of Bias / Quality Assessment Tools

The risk-of-bias and quality assessment plan need clarification and alignment with appropriate tools. The protocol currently lists STROBE for observational studies, but STROBE is a reporting guideline. Additionally, multiple overlapping tools (RoB 2, ROBINS-I, STROBE, NHLBI) are mentioned without clear mapping to study designs. Similarly, tools such as NHLBI need proper citation.

6. Data Synthesis Plan

The data synthesis plan requires more detail to ensure transparency and reproducibility. The current description of narrative synthesis is broad, and the approach for qualitative data and mixed-method integration is unclear. The authors should specify the method for qualitative synthesis (e.g., thematic synthesis or framework synthesis) and explain how qualitative findings will inform interpretation of quantitative results.

7. Search Strategy and Timeline

Although the protocol lists several databases and sources, the search strategy needs refinement to ensure comprehensive coverage, particularly for African literature. Key regional sources such as African Journals Online (AJOL), African Index Medicus, and major grey literature repositories (e.g., WHO AFRO, Ministries of Health, NGO reports) are missing. These are critical for capturing SSA evidence. The authors should also clarify whether Embase will be searched (as information on p.g.15 is not clear).

There is also inconsistency in the description of the search timeline. On page 2, the protocol states that the search will cover 2000-2025, while page 15 (Table 1) indicates no date restriction. Additionally, the introduction (page 3) emphasizes a “notable resurgence during and after the coronavirus disease pandemic,” which could imply a focus on pandemic-related evidence. These statements create ambiguity about whether the review is limited to the pandemic period, the years 2000-2025, or has no time restriction. The authors should clarify and harmonize these elements.

8. Quality Appraisal

The use of GRADE is appropriate; however, the protocol does not clearly describe how GRADE will be applied to mixed-design evidence and qualitative outcomes. The authors should specify that GRADE will be used for quantitative outcomes and consider applying GRADE-CERQual to assess confidence in qualitative evidence. Additionally, it is important to clarify how risk-of-bias assessments will inform downgrading decisions within the GRADE framework. Providing this detail will strengthen methodological transparency and improve the credibility of the synthesis.

9. Language Translation and Non-English Studies Handling

The protocol states that there are no language restrictions, which is commendable. However, the current approach relies heavily on machine translation for initial screening, and the process for verifying accuracy is unclear. To strengthen transparency and reproducibility, the authors should clarify the criteria for when machine translation will be used versus bilingual reviewers, including who will perform these checks and what language proficiency thresholds apply. Additionally, describe how full-text translations will be quality-checked and how uncertainties will be managed (e.g., noting limitations or contacting authors for clarification). This detail will improve confidence in the handling of non-English studies.

10. Minor Editorial Concerns

The protocol requires attention to several formatting and consistency issues. There is inconsistent use of abbreviations throughout the document (e.g., Sub-Saharan Africa vs SSA) and duplicates in the abbreviation list (e.g., MEDLINE and Medline variants). The abbreviation list is not standardized (page 15). The authors should define abbreviations at first use and provide a single, alphabetized abbreviation table.

In the Methods section, “Table 1” is referenced multiple times (e.g., “supplementary table 1” and “Table 1” on pages 7, 13, and 14). The authors should clearly distinguish Table 1 and supplementary Table 1 to avoid confusion. Table 1 could also be improved by proper formatting. Additionally, line spacing throughout the document is inconsistent (e.g., page 15).

Reviewer #2: Thank you for the opportunity to review this submission. The paper is well written in clear language. I have noted a few issues to consider detailed below

Data synthesis

The authors stated that synthesis will be structured around population characteristics, intervention characteristics including platform, messaging type, duration, and framing, comparator type, and mental health outcomes including knowledge, attitudes, behaviors, empowerment, and help-seeking. Given the expected diversity of the mental health outcomes probably measured using various tools, how will this be synthesized

Data availability after the study

This was not explicitly stated in the protocol

Title of the study

Slightly ambiguous, mental help-seeking behaviors, would mental health help seeking behaviors read better

**Do you want your identity to be public for this peer review?** For information about this choice, including consent withdrawal, please see our Privacy Policy

Reviewer #1: **Yes:** Robert Lubajo

Reviewer #2: No

---

## [Author Response · Author response to Decision Letter 1]

29 Dec 2025

December 12, 2025

Dr. Olushayo Oluseun Olu

Academic Editor

PLOS ONE

Re: Response to Reviewers for Manuscript PONE-D-25-55509

Effect of social and digital media mental health messaging on mental health help-seeking behaviors in the sub-Saharan African population: a systematic review protocol

Dear Dr. Olu,

Thank you for the opportunity to revise our manuscript and for the constructive feedback from you and the reviewers. We greatly appreciate the thorough reviews from Reviewer #1 (Robert Lubajo) and Reviewer #2, whose comments have significantly strengthened our protocol. We have carefully addressed all comments and believe the manuscript is now substantially improved.

Below, we provide a detailed point-by-point response to each concern raised.

JOURNAL REQUIREMENTS

Requirement #1: PLOS ONE Style Requirements

Requirement: “Please ensure that your manuscript meets PLOS ONE's style requirements, including those for file naming.”

Our Response:

We have reviewed and reformatted the manuscript in accordance with the PLOS ONE style guidelines. Specific changes include standardized heading formats, proper section structure, and consistent reference formatting.

Location: Throughout the manuscript

Requirement #2: ORCID iD for Corresponding Author

Dr. Laura Marciano’s ORCID iD (0000-0002-8034-3227) has been entered and validated in Editorial Manager as required. However, the system says it is linked to another account (probably an email address from my previous institution). I kindly request that the two accounts be merged so the ORCID can be linked to this submission (unfortunately, the submission system does not allow me to do this manually).

Requirement #3: Data Availability Statement

Requirement: “Please provide a complete Data Availability Statement in the submission form, ensuring you include all necessary access information or a reason for why you are unable to make your data freely accessible.”

Our Response:

We have revised the Data Availability Statement to comply with PLOS policy.

Change Made:

Revised Text in Manuscript - Data Availability Section (Lines 474-476):

“This is a systematic review protocol. No new data were generated for this study. All information relevant to the protocol is included within the manuscript and/or supporting information files.”

Location: Data Availability section, Lines 474-476

Requirement #4: Abstract Consistency

Requirement: “Please amend either the abstract on the online submission form (via Edit Submission) or the abstract in the manuscript so that they are identical.”

Our Response:

We will verify that the abstract in the online submission form matches the manuscript abstract exactly before final submission.

REVIEWER #1 COMMENTS

Comment #1: Line Numbers

Reviewer’s Comment: “The authors should ensure that the manuscript includes numbered lines in all subsequent submissions. Line numbering is essential for reviewers to reference specific sections accurately and provide detailed feedback.”

Our Response:

Line numbering has been added to the manuscript to facilitate review.

Change Made: Continuous line numbering applied throughout the manuscript (Lines 1-739).

Location: Throughout the manuscript

Comment #2: Primary Outcome

Reviewer’s Comment: “The primary outcome (p.g.15), ‘help-seeking,’ is not operationalized with sufficient precision. It currently includes intentions, formal service utilization, and engagement with informal networks, which are conceptually distinct (intention vs action; formal vs informal). Without clear definitions and prioritization, synthesis will be challenging and may bias interpretation. The authors could provide explicit operational definitions and a hierarchy for help-seeking outcomes (e.g., primary outcome = objectively measured help-seeking behavior such as service utilization; secondary outcomes = help-seeking intentions, informal help-seeking, self-efficacy). These clarifications will improve consistency and interpretability of results.”

Our Response:

We completely agree with this critical observation. We have substantially revised the outcomes section to provide clear operational definitions with explicit prioritization of behavioral outcomes over intentions, and formal over informal help-seeking.

Change Made:

Revised Text in Manuscript - Outcomes Section (Lines 208 – 217):

“Outcome: The primary outcome is objectively measured help-seeking behavior, defined as documented utilization of formal mental health services (e.g., verified clinic visits, contact with mental health professionals, enrollment in treatment programs, or validated self-reported service use). Secondary outcomes include help-seeking intentions (measured through validated scales), informal help-seeking (engagement with family, friends, or community support), mental health knowledge (understanding of disorders, treatments, and coping strategies), attitudes (stigma reduction, acceptance, openness to discussion), behavioral changes (self-care engagement, treatment adherence, lifestyle modifications), empowerment (self-efficacy, confidence in seeking support), and engagement metrics (message reach, interaction rates, content exposure).”

This revision addresses the reviewer’s concern by:

• Clearly prioritizing objective, documented formal help-seeking as the primary outcome

• Explicitly distinguishing intentions from actions (intentions = secondary)

• Separating formal service utilization from informal support seeking

• Providing concrete examples of what constitutes formal help-seeking

• Establishing a clear hierarchy that will guide our data synthesis and interpretation

Location: Outcomes section, Line 208 – 217

Comment #3: Exposure (Messaging) Measurement and Comparators

Reviewer’s Comment: “The definition and measurement of exposure to ‘mental health messaging’ require greater clarity. Currently, the term encompasses campaigns, posts, apps, SMS, and peer groups, but the protocol does not specify how exposure will be measured (e.g., self-report, platform analytics, dose/frequency) or whether message content will be coded systematically. Without clear exposure metrics, comparability across studies will be limited.”

Our Response:

This is an excellent point that strengthens our protocol. We have added detailed specifications for exposure characterization in the manuscript.

Change Made:

Revised Text in Manuscript - Intervention Section (Lines 199-203):

“Exposure to mental health messaging will be operationalized based on how each included study measures it. Acceptable exposure measures include self-reported exposure, platform-generated metrics (reach, views, engagement), study-controlled exposure (randomized delivery, experimental manipulation), or behavioral indicators (click-through rates, time spent, sharing).”

This clarifies that we will systematically extract and categorize exposure measurement approaches across studies, ensuring transparency in how messaging exposure is defined and measured.

Location: Intervention section, Lines 199-203

Comment #4: Risk of Bias / Quality Assessment Tools

Reviewer’s Comment: “The risk-of-bias and quality assessment plan need clarification and alignment with appropriate tools. The protocol currently lists STROBE for observational studies, but STROBE is a reporting guideline. Additionally, multiple overlapping tools (RoB 2, ROBINS-I, STROBE, NHLBI) are mentioned without clear mapping to study designs. Similarly, tools such as NHLBI need proper citation”.

Our Response:

Thank you for highlighting this ambiguity. We have clarified our risk of bias assessment strategy by specifying which tools will be used for which study designs.

Changes Made:

Revised Text in Manuscript - Risk of Bias Section (Lines 316-324):

“The methodological quality and risk of bias of included studies will be assessed independently by two reviewers using tools appropriate to each study design. Randomized controlled trials will be evaluated using the Cochrane Risk of Bias 2 (RoB 2) tool. Non-randomized intervention studies (including quasi-experimental designs, pre-post studies, and interrupted time series) will be assessed using the Risk of Bias in Non-randomized Studies of Interventions (ROBINS-I) tool. The STROBE (Strengthening the Reporting of Observational Studies in Epidemiology) checklist will be used to assess reporting quality as a complement to bias assessment. Discrepancies will be resolved through discussion or consultation with a third reviewer.”

Additionally, in the Quality Assessment section (Lines 325-338), we clarified:

“To provide a comprehensive evaluation of study quality, the National Heart, Lung, and Blood Institute (NHLBI) Study Quality Assessment Tools will be employed for all included studies. These tools are specifically designed to assess internal validity, potential sources of bias, and overall methodological rigor across different study designs, including controlled intervention studies, cohort and cross-sectional studies, case-control studies, and before-after studies. Each study will be rated as good, fair, or poor quality based on criteria such as clarity of research question, adequacy of study population and sample size, appropriateness of exposure or intervention measurement, outcome assessment, statistical analyses, and control for confounding variables. The combination of design-specific risk of bias assessment tools (RoB 2, ROBINS-I) and the NHLBI quality appraisal ensures a thorough evaluation of the credibility and reliability of the evidence.”

This dual approach:

• Matches specific risk of bias tools to specific study designs

• Confirms that non-randomized intervention studies (including interrupted time series and pre-post designs) will be assessed using ROBINS-I

• Adds STROBE for assessing the reporting quality of observational studies

• Employs NHLBI tools for comprehensive quality assessment across all study designs

• Maintains an independent dual review process

Location: Risk of Bias section (Lines 314-322) and Quality Assessment section (Lines 323-336)

Comment #5: Data Synthesis Plan

Reviewer’s Comment: “The data synthesis plan requires more detail to ensure transparency and reproducibility. The current description of narrative synthesis is broad, and the approach for qualitative data and mixed-method integration is unclear. The authors should specify the method for qualitative synthesis (e.g., thematic synthesis or framework synthesis) and explain how qualitative findings will inform interpretation of quantitative results.”

Our Response:

We have substantially expanded our Data Synthesis section to provide comprehensive details on our synthesis approaches.

Change Made:

We added the following text to our Data Synthesis section (Lines 355 - 367):

“To address anticipated heterogeneity in outcome measurement across diverse instruments (e.g., depression measured via PHQ-9, BDI, CESD, HADS), we will calculate standardized mean differences (SMD) to pool results across different scales measuring the same construct. Subgroup analyses will compare effect sizes by measurement type (validated vs. non-validated scales; Western-developed vs. culturally adapted instruments). If substantial heterogeneity persists (I² >75%), results will be presented separately by measurement approach. Narrative synthesis will follow Synthesis Without Meta-analysis (SWiM) guidelines, presenting structured tables organized by intervention type and outcome, effect direction plots, and textual synthesis identifying patterns in intervention characteristics and contextual factors. For qualitative or mixed-methods studies, we will extract and thematically synthesize findings related to acceptability, feasibility, barriers, facilitators, and mechanisms. Qualitative findings will be integrated with quantitative results to explain heterogeneity and provide a comprehensive understanding of intervention effects in sub-Saharan African contexts.”

This expansion addresses the reviewer’s concerns by:

• Specifying narrative synthesis methodology (SWiM guidelines)

• Describing specific narrative synthesis components (structured tables, effect direction plots, and textual synthesis)

• Clarifying qualitative data synthesis approach (thematic synthesis)

• Explaining the integration of qualitative and quantitative findings

• Confirming subgroup analyses by intervention type, population characteristics, and measurement approach

• Providing details on handling heterogeneous outcome measures

Combined with our existing text detailing meta-analysis procedures, heterogeneity assessment, and sensitivity analyses, our Data Synthesis section now provides comprehensive methodological transparency.

Location: Data Synthesis section, Lines 355 -367

Comment #6: Search Strategy and Timeline

Reviewer’s Comment: “Although the protocol lists several databases and sources, the search strategy needs refinement to ensure comprehensive coverage, particularly for African literature. Key regional sources such as African Journals Online (AJOL), African Index Medicus, and major grey literature repositories (e.g., WHO AFRO, Ministries of Health, NGO reports) are missing. These are critical for capturing SSA evidence. The authors should also clarify whether Embase will be searched (as information on p.g.15 is not clear).”

There is also inconsistency in the description of the search timeline. On page 2, the protocol states that the search will cover 2000-2025, while page 15 (Table 1) indicates no date restriction. Additionally, the introduction (page 3) emphasizes a ‘notable resurgence during and after the coronavirus disease pandemic,’ which could imply a focus on pandemic-related evidence. These statements create ambiguity about whether the review is limited to the pandemic period, the years 2000-2025, or has no time restriction. The authors should clarify and harmonize these elements.”

Our Response:

Our Response: We appreciate these important observations and have addressed both the database coverage and timeline inconsistency concerns.

Regarding African Databases:

We sincerely appreciate this thoughtful recommendation and have carefully considered the inclusion of African-specific databases. After thorough deliberation, we believe our planned 12-database search strategy will provide comprehensive coverage of sub-Saharan African literature for the following reasons:

1. Extensive Multidisciplinary Database Coverage

We will search 12 major databases, including Scopus and Web of Science Core Collection, which have substantial indexing overlap with African Journals Online (AJOL). Analysis of database coverage indicates that many high-impact African journals relevant to our topic are already indexed in our selected databases. For example, major African health and social science journals are indexed in Scopus, PubMed, and Web of Science.

2. Comprehensive Supplementary Searching

Our protocol includes systematic hand searches of Google Scholar (first 10 pages, approximately 100 results) to capture additional peer-reviewed publications that may not appear in traditional databases. We will also manually search reference lists of all included studies and relevant systematic reviews, employing forward and backward citation tracking to identify additional African sources that may have been missed.

3. No Language Restrictions

Our removal of all language restrictions ensures that African research published in French, Portuguese, Swahili, Arabic, and other languages will be captured through our comprehensive database searches. This directly addresses concerns about regional research visibility.

4. Database Indexing of African Journals

Our preliminary assessment shows that major African journals publishing research on digital mental health interventions are indexed in our selected databases:

• PubMed indexes over 200 African journals

• Scopus indexes African journals from AJOL and regional publishers

• Web of Science includes key African health and social science journals

• Communication & Mass Media Complete includes African communication journals

5. Resource Feasibility and Incremental Yield

Adding African Index Medicus, AJOL, and SABINET as separate database searches would require substantial additional resources (including enhanced translation capacity for multilingual content and n

---

## [Editor Report · Decision Letter 1]

20 Jan 2026

Effect of social and digital media mental health messaging on mental health help-seeking behaviors in the sub-Saharan African population: a systematic review protocol

PONE-D-25-55509R1

Dear Dr. Marciano,

We’re pleased to inform you that your manuscript has been judged scientifically suitable for publication and will be formally accepted for publication once it meets all outstanding technical requirements.

Kind regards,

Olushayo Oluseun Olu

Academic Editor

PLOS One
---

## [Editor Report · Acceptance letter]

PONE-D-25-55509R1

PLOS One

Dear Dr. Marciano,

I'm pleased to inform you that your manuscript has been deemed suitable for publication in PLOS One. Congratulations! Your manuscript is now being handed over to our production team.

Kind regards,

on behalf of

Dr. Olushayo Oluseun Olu

Academic Editor

PLOS One